# Overproduction of Human Zip (SLC39) Zinc Transporters in *Saccharomyces cerevisiae* for Biophysical Characterization

**DOI:** 10.3390/cells10020213

**Published:** 2021-01-21

**Authors:** Eva Ramos Becares, Per Amstrup Pedersen, Pontus Gourdon, Kamil Gotfryd

**Affiliations:** 1Membrane Protein Structural Biology Group, Department of Biomedical Sciences, Faculty of Health and Medical Sciences, University of Copenhagen, Maersk Tower 7-9, DK-2200 Copenhagen N, Denmark; evabecares@sund.ku.dk; 2Department of Biology, Faculty of Science, University of Copenhagen, Universitetsparken 13, DK-2100 Copenhagen OE, Denmark; papedersen@bio.ku.dk; 3Department of Experimental Medical Science, Lund University, Sölvegatan 19, SE-221 84 Lund, Sweden

**Keywords:** membrane proteins, overproduction, production platform, protein purification, *Saccharomyces cerevisiae*, solute carrier 39, SLC39, family, yeast, zinc, zinc transporters, ZIPs

## Abstract

Zinc constitutes the second most abundant transition metal in the human body, and it is implicated in numerous cellular processes, including cell division, DNA and protein synthesis as well as for the catalytic activity of many enzymes. Two major membrane protein families facilitate zinc homeostasis in the animal kingdom, i.e., Zrt/Irt-like proteins (ZIPs aka solute carrier 39, SLC39, family) and Zn transporters (ZnTs), essentially conducting zinc flux in the opposite directions. Human ZIPs (hZIPs) regulate import of extracellular zinc to the cytosol, being critical in preventing overaccumulation of this potentially toxic metal, and crucial for diverse physiological and pathological processes, including development of neurodegenerative disorders and several cancers. To date, our understanding of structure–function relationships governing hZIP-mediated zinc transport mechanism is scarce, mainly due to the notorious difficulty in overproduction of these proteins for biophysical characterization. Here we describe employment of a *Saccharomyces cerevisiae*-based platform for heterologous expression of hZIPs. We demonstrate that yeast is able to produce four full-length hZIP members belonging to three different subfamilies. One target (hZIP1) is purified in the high quantity and homogeneity required for the downstream biochemical analysis. Our work demonstrates the potential of the described production system for future structural and functional studies of hZIP transporters.

## 1. Introduction

Zinc is one of the most crucial minerals for both plants and animals, and constitutes the second most abundant transition metal in humans. Up to 10% of the human proteome associates with zinc, and this metal exerts essential structural and/or functional roles, enabling correct activity of many proteins [1]. Notably, zinc is critical for many transcription factors and serves as a cofactor in all enzyme classes. In addition, it acts as a signaling molecule, and abnormalities of intracellular levels of zinc affect pathways controlling cell development, growth, differentiation, and death [2]. 

Since zinc is essential and yet toxic in excess, both body and cellular concentrations of this trace element have to be tightly regulated. In the animal kingdom, zinc transport is primarily achieved by membrane proteins (MPs) belonging to two solute carrier (SLC) families, i.e., SLC30 and SLC39. ZnT transporters (SLC30 family) reduce cytoplasmic zinc levels by providing cellular efflux or uptake into intracellular compartments. Conversely, Zrt-, Irt-like proteins (ZIPs) of the SLC39 family increase the intracellular concentration of the metal by mediating cellular uptake or efflux from intracellular compartments [3,4,5,6,7]. 

In humans, the ZIP family encompasses 14 members that, based on sequence similarity, are divided into four subfamilies, i.e., hZIP I, hZIP II, GufA, and LIV-1 (Figure 1A) [8,9]. Human ZIPs (hZIPs) have been proposed to regulate zinc homeostasis in a tissue- and cell type-specific manner, as reflected by their subcellular localization [10,11,12,13]. As examples, hZIP1 mediates microglial activation by autocrine and paracrine signaling [14]; hZIP2 is specifically expressed in the epidermis, responsible for differentiation of keratinocytes and skin turnover [15]; hZIP4 is located to the apical membrane of enterocytes and regulates dietary zinc absorption [16]; whereas hZIP10 plays an important role in maintenance of the immune system, as it is key for early β-cell survival [17]. In addition, altered protein expression or malfunction of hZIPs has been linked to a plethora of conditions, including several cancers, dermatitis, and neurodegenerative autoimmune diseases [6]. For instance, reduced levels of hZIP1, 2, and 3 were reported in the two types of prostate cancers [18,19], and upregulation of hZIP4 and 6 was shown to stimulate proliferation of pancreas and breast tumors, respectively [20,21]. Moreover, hZIP11 single nucleotide polymorphisms correlate with the development of bladder cancer, having either protective or increased risk of cancer progression [22]. Furthermore, loss-of-function mutations in hZIP4 lead to acrodermatitis enteropathica, a rare genetic disorder leading to Zn^2+^ deficiency manifested by skin lesions, growth retardation, immune system dysfunction, and neurological complications [23]. In addition, hZIP13 has been demonstrated to play a key role in development of the connective tissue and its malfunction can result in spondylocheiro dysplastic form of Ehlers-Danlos syndrome, a rare disorder of the connective tissue [24,25]. Consequently, hZIPs represent attractive targets to discover novel compounds for therapeutic purposes. 

Despite the crucial role of hZIPs in health and disease, the understanding of this family of transporters remains sparse, as reflected by lack of structural and mechanistic information. Based on sequence analyses, hZIPs are predicted to consist of eight transmembrane segments (TMs), with N- and C-termini facing the noncytosolic space (Figure 1B) [13], of which the N-termini vary considerably in length (Figure 2A). Moreover, ZIPs usually form homodimers, but hZIP6/hZIP13 heterodimers have also been reported [10]. The only available structures are the inward-open metal-bound state of a full-length ZIP originate from a prokaryotic homologue, i.e., *Bordetella bronchiseptica* (BbZIP). These data indicate suggested that TM2, 4, 5, and 7 form a central helical core containing a binuclear metal binding center, while the remaining TMs (i.e., TM1, 3, 6, and 8) are located more peripherally (Figure 1C) [26,27]. In addition, the BbZIP structure indicated that the eight TMs are grouped into two units, one including TM1, 4, 5, and 6, while the second consists of the other four TMs (Figure 1C), forming the binuclear metal binding center in-between [27]. For hZIPs the available structural information is limited to the structure of the extracellular N-terminal domain of hZIP4 [28], a stretch that has been shown to play a critical role in dimerization, based on a highly conserved PAL motif [28]. Nevertheless, the function and transport mechanism of hZIPs remain largely elusive, and in addition, functional data suggest that ZIP members can bind different metals with varying specificities, affinities, and kinetics [29], implicating hitherto poorly understood protein-specific adaptations of the zinc transport. Thus, complementary structural information of other ZIP members and conformations are required to understand the molecular determinants governing the zinc transfer.

As for other MPs, structural investigation of ZIPs is obstructed by the difficulty in obtaining protein samples of sufficient quantity and quality. This is highlighted by the fact that ZIP transporters have been exceptionally difficult to overproduce and purify, perhaps related to their N_out_-C_out_ topology [30,31]. In fact, BbZIP is the only ZIP member that has been successfully isolated to homogeneity in solution to date, although over 50 ZIP homologs have been attempted [31,32]. Moreover, even BbZIP was reported to be stable only in the presence of Cd^2+^, resulting in tightly bound ions in the binuclear metal binding center [27]. Among the 14 hZIPs, only hZIP13 has been successfully overexpressed in sufficient quantities to perform biochemical characterization, through implementation of an insect Sf9 cell-based expression system, but no information on obtained protein yields is available [24].

Traditionally, due to the availability of expression plasmids, its easy genetic manipulation, high growth rate, and low cost, *Escherichia coli* has been the primary choice as an expression system for delivery of MPs [33,34]. However, *E. coli*-based platforms display major limitations when expressing proteins from higher organisms, where often expression systems of eukaryotic origin, such as yeast, insect, or mammalian cell-derived, are required [35]. Compared to *E. coli*, establishment of insect- or mammalian-based heterologous expression platforms can be cumbersome, tedious, and expensive. In contrast, yeast systems offer a cheap, easy, and robust large-scale production, combined with the ability to perform some of the post-translational modifications required for proper protein folding [36,37,38]. Hence, yeast represents an attractive complementary host for high-quality MP overproduction, permitting downstream structure–function characterization [39,40]. 

The *Saccharomyces cerevisiae* strain PAP1500 combined with a high copy number expression vector containing the CYC-GAL promoter (CG-P) has been successfully exploited for production of diverse classes of human MPs [39,41,42,43]. In this tandem system, the Galactose 4 (Gal4) transcription activator is overexpressed in the host upon galactose supplementation to the media, and the activator is employed for induction of the CYC-GAL (CG-P) promoter present in the expression plasmid [44]. The *URA3* and *leu2-d* selection markers are also included in the vector to enable easy selection of the transformants, and to ensure high amounts of plasmid prior to induction in a uracil- and leucine-deficient media [45].

Here we report development of a cost-effective and efficient method to express hZIPs, employing the PAP1500-based MP production system. Briefly, we attempted expression of four selected hZIPs, i.e., hZIP1, 2, 11, and 13, belonging to three different subfamilies, and produced them as full-length versions with C-terminal green fluorescent protein (GFP) fusions. Subsequently, utilizing GFP florescence, we screened to select the most promising target for large-scale production, both by analyzing expression levels and by identifying suitable detergent(s) for membrane extraction using florescence-detection size-exclusion chromatography (F-SEC). For the most promising candidate, i.e., hZIP1, we proceeded with affinity chromatography-based protein purification. We also refined the procedure through investigation of how different tags and their localization affect hZIP1 production and solubilization. Collectively, our results demonstrate that *S. cerevisiae* is capable of producing hZIPs in large scale for downstream biophysical characterization.

## 2. Materials and Methods 

### 2.1. Engineering of Expression Plasmids

cDNAs encoding full-length hZIPs were commercially synthetized by GenScript (New Jersey, NJ, USA). All genes were codon-optimized for *Saccharomyces cerevisiae*-based expression using OptimumGene^TM^ algorithm, including adjustment of parameters crucial for transcription, translation, and protein folding. A panel of constructs was designed to screen for the levels of expression and to enable protein purification (Figure 2A). For the initial screening, engineered constructs included a C-terminally fused tag containing tobacco etch virus (TEV) protease cleavage site (ENLYFQ↓SQF), GFP, and a His_8_ stretch (denoted hZIP-TEV-GFP-His throughout). Subsequently, optimization of production of the selected target, i.e., hZIP1, was performed with GFP-free constructs, including only either N- or C-terminal TEV protease-detachable purification tag enabling affinity chromatography step (His_8_- or StrepII-tag). These comprised the following two variants, i.e., His-TEV-hZIP1 and StrepII-TEV-hZIP1, respectively. All expression constructs were generated by PCR amplification of the respective hZIP cDNAs with or without GFP fragment using AccuPol DNA polymerase (Amplicon, Odense, Denmark) and the primers listed in Appendix A. Subsequently, each expression plasmid was assembled employing homologous recombination directly in the *S. cerevisiae* strain PAP1500 (α *ura3*-*52 trp1::GAL10-GAL4 lys2*-*801 leu2*Δ*1 his3*Δ*200 pep4::HIS3 prb1*Δ*1.6R can1 GAL*; Figure 2C) [47] with the corresponding hZIP/GFP PCR fragments and a *Bam*HI-, *Hin*dIII- and *Sal*I-digested pEMBLyex4 vector [44]. Transformants were selected on minimal medium agar plates supplemented with glucose (2 g L^−1^), leucine (60 mg L^−1^), and lysine (30 mg L^−1^). DNA sequencing was performed on isolated plasmids to verify their identity.

### 2.2. Overproduction of hZIPs, Live-Cell Bioimaging and In-Gel GFP Fluorescence

All hZIPs were produced using the *S. cerevisiae* expression strain PAP1500 (Figure 2C) essentially as previously reported [47]. For screening purposes, expression of hZIP-TEV-GFP-His, His-TEV-hZIP1, and StrepII-TEV-hZIP1 constructs was performed in 2-L cell culture scale using shaker flasks. Briefly, a single colony of each transformant was used to inoculate 5 mL of minimal media containing glucose (2 g L^−1^), leucine (60 mg L^−1^), and lysine (30 mg L^−1^), and grown overnight at 30 °C until OD_450_ ranged from 0.5 to 1.0. Subsequently, 0.5 mL of the preculture was transferred to 5 mL of the minimal medium w/o leucine and grown for another 24 h at 30 °C to increase plasmid copy number. The following day, the culture was scaled up to 100 mL in the same media for an additional 24 h and used to inoculate 2 L of media containing amino acids (alanine (20 mg L^−1^), arginine (20 mg L^−1^), aspartic acid (100 mg L^−1^), cysteine (20 mg L^−1^), glutamic acid (100 mg L^−1^), histidine (20 mg L^−1^), lysine (30 mg L^−1^), methionine (20 mg L^−1^), phenylalanine (50 mg L^−1^), proline (20 mg L^−1^), serine (375 mg L^−1^), threonine (200 mg L^−1^), tryptophan (20 mg L^−1^), tyrosine (30 mg L^−1^) and valine (150 mg L^−1^)), glucose (10 g L^−1^) and glycerol (3%, *v*/*v*). Upon consumption of sugar, protein expression was induced by supplementing the culture with galactose (final concentration of 2%) and allowed for 48–72 h at 15 °C. Subsequently, cells were harvested (5800× *g*, 15 min, 4 °C) and stored at −80 °C. Obtained material typically yielded ≈3–4 g of wet cell pellet per L of cell culture. For initial purification of hZIP1-TEV-GFP-His, cell material obtained from 2-L yeast culture induced for expression for 72 h was used. Overproduction of His-TEV-hZIP1 and StrepII-TEV-hZIP1 constructs for large-scale purification was performed identically, but in 12-L scale with 68-h induction. 

Localization of TEV-GFP-His hZIP fusions was performed by live-cell bioimaging where GFP fluorescence was analyzed using the Nikon Eclipse E600 microscope (Nikon, Tokyo, Japan) equipped with an Optronics MagnaFire S99802 camera (Optronics, Muskogee, OK, USA).

In-gel GFP fluorescence was detected on samples resolved by 4–20% SDS-PAGE (Thermo Fisher Scientific, Waltham, MA, USA), where the signal was immediately visualized with ImageQuant LAS 4000 imaging system and the corresponding control software (both from GE Healthcare, Copenhagen, Denmark).

### 2.3. Preparation of Crude Membranes

For initial screening purposes of hZIP-TEV-GFP-His expression and purification, wet cell pellets were resuspended in ice-cold lysis buffer (50 mM Tris-HCl pH 8.0, 200 mM NaCl and 30% glycerol) supplemented with 5 mM β-mercaptoethanol (BME), 1 mM PMSF, and SigmaFAST^TM^ protease inhibitor cocktail (Sigma-Aldrich, St. Louis Missouri, MO, USA). Subsequently, cell suspension was mixed with glass beads and mechanically ruptured using a bead beater (BioSpec, Bartlesville, OK, USA). Following cell disruption, the supernatant was collected, and glass beads were washed in ice-cold lysis buffer. Unbroken cells and cell debris were removed from the homogenate by centrifugation (1000 or 3000× *g*, 10 min, 4 °C). Crude membranes were then pelleted by ultracentrifugation (138,000× *g*, 3 h, 4 °C), resuspended in the above-mentioned buffer (see Appendix A) using a Potter–Elvehjem homogenizer and stored at −80 °C.

In contrast, cells overexpressing both N-terminal hZIP1 fusions, i.e., His-TEV-hZIP1 and StrepII-TEV-hZIP1, were disrupted by manual bead beating. Briefly, for His-TEV-hZIP1, the wet cell pellet obtained from 12-L culture (typically yielding ≈80 g) was resuspended in ice-cold lysis buffer (25 mM Tris-HCl pH 7.0, 500 mM NaCl, and 20% glycerol) supplemented with 250 µM EDTA, 250 µM EGTA, 5 mM BME, 1 mM PMSF, 1 μg mL^−1^ chymostatin, 1 μg mL^−1^ leupeptin, and 1 μg mL^−1^ pepstatin. Cells expressing StrepII-TEV-hZIP1 were lysed identically, but in the buffer with pH of 8.0 (25 mM Tris-HCl). Subsequently, 12.5 mL of the cell suspension was mixed with 15 mL of glass beads directly in 50-mL conical tubes and the mixture was homogenized by high-speed vortexing (eight cycles of 1 min with 1-min cooling in between). The resulting supernatant was collected and treated identically as described above for the C-terminally fused constructs. Following ultracentrifugation, crude membranes were resuspended in the respective solubilization buffers containing all the supplements present in the lysis buffer mentioned above (see Appendix A).

### 2.4. Detergent Screening, Immunoblotting, and F-SEC Analysis

Following estimation of the total protein concentration using Bradford assay [48] (Sigma-Aldrich), crude membranes were diluted in the corresponding solubilization buffers to a final concentration of 2–5 mg L^−1^. Subsequently, small-scale solubilization screens were performed using 0.5 mL of the respective membranes vigorously rotated (90–120 min, 4 °C) in the presence of the below-mentioned detergents. hZIP1-, hZIP2-, and hZIP13-TEV-GFP-His-expressing membranes were solubilized with a final concentration of 1% (*w*/*v*) of n-dodecylphosphocholine (FC-12), n-decyl-D-maltoside (DM), or n-dodecyl-D-maltoside (DDM) with or without 0.34% (*w*/*v*) of cholesteryl hemisuccinate Tris salt (CHS); all from Anatrace, Maumee, OH, USA. Solubilization efficacy of hZIP11-TEV-GFP-His was tested with a double final concentration of the above-mentioned surfactants, i.e., 2% of detergent ±0.68% of CHS. Extraction of His-TEV-hZIP1 was performed only with DDM and CHS tested at three ratios (1% + 0.1%, 2% + 0.2% and 5% + 0.5%, *w*/*v*, respectively). Subsequently, insoluble material was removed by ultracentrifugation (50,000× *g*, 20 min, 4 °C). Supernatant originating from the respective TEV-GFP-His hZIP fusions was used directly to measure GFP fluorescence (excitation 485 nm, emission 520 nm) to assess detergent extraction efficacy and/or for SDS-PAGE analysis. Extracted fractions of His-TEV- and StrepII-TEV-hZIP1 were evaluated by immunoblotting. Briefly, the respective samples were resolved by 4–20% SDS-PAGE and transferred to PVDF membranes (0.45 μm; both from Thermo Fisher Scientific). Following blocking, membranes were probed with 6×His mAb-HRP conjugate (Takara Bio, Mountain View, CA, USA) or Strep-Tactin^®^ HRP conjugate (IBA GmbH, Göttingen, Germany) according to recommendations of the manufacturers. Immunoblots were visualized by chemiluminescence with SuperSignal^TM^ West Femto Maximum Sensitivity substrate (Thermo Fisher Scientific) using AlphaImager^TM^ (Alpha Innotech, San Leandro, CA, USA). Relative signal intensities of the bands were quantified using ImageJ software (https://imagej.nih.gov/ij/).

In addition, extracted TEV-GFP-His hZIP fusions were analyzed using florescence-detection size-exclusion chromatography (F-SEC). Briefly, solubilized material was analyzed on a Superdex 200 Increase 10/300 GL column (GE Healthcare) equilibrated with F-SEC buffer (20 mM Tris-HCl pH 8, 100 mM NaCl, 2 mM BME, 10% glycerol and 0.03% DDM; Anatrace) attached to ÄKTA Pure system (GE Healthcare) equipped with a Prominence RF-20A fluorescence detector (Shimadzu, Kyoto, Japan).

### 2.5. Purification of hZIP1, TEV Protease Cleavage, and SEC Analysis

Purification of the two His-tagged hZIP1 variants, i.e., hZIP1-TEV-GFP-His and His-TEV-hZIP1 was performed using immobilized metal ion affinity chromatography (IMAC), whereas StrepII-TEV-hZIP1 was purified employing StrepII-tag affinity chromatography. In all cases, the starting material included crude membranes isolated from flask-grown yeast cells (for the corresponding culture conditions see Section 2.3). Details about solubilization conditions and chromatography buffers are presented in Appendix A. Following solubilization at 4 °C, insoluble material was removed by ultracentrifugation (80,000× *g*, 3 h, 4 °C) and subjected to IMAC-based purification. Briefly, solubilized His-tagged hZIP1 variants were diluted in the corresponding solubilization buffers (Appendix A) to reduce detergent concentration and loaded onto a 5-mL HisTrap HP columns attached to an Äkta Pure system (both from GE Healthcare). Following column wash, bound proteins were eluted in IMAC buffers using a linear imidazole gradient (25/50–500 mM; Appendix A). Subsequently, top IMAC fractions were pooled, concentrated on Vivaspin concentrators (MWCO 30 or 50 kDa; Sartorius, Göttingen, Germany) and obtained His-tagged hZIP1 samples were cleaved with TEV-His_10_-tagged protease (home source; 16 h, 4 °C) used in TEV-to-hZIP1 ratio of 1:10 (*w*/*w*) in MWCO 10 kDa dialysis bags (Thermo Fisher Scientific) with concomitant dialysis against IMAC buffer containing 20 mM imidazole. Subsequently, reverse (R)-IMAC was performed to separate cleaved, i.e., TEV-GFP-His- or His-TEV-free hZIP1 from uncleaved fraction, TEV protease and released fusion tags, all containing His-tag and rebinding to the IMAC resin. TEV protease cleavage products were analyzed by SDS-PAGE and visualized by in-gel GFP fluorescence or immunoblotting. 

Solubilized StrepII-TEV-hZIP1 membranes were also diluted to reduce detergent concentration and the pH of the solubilizate was adjusted to reach a final value of 8.0, prior to loading onto a 5-mL Strep-Tactin^®^XT Superflow^®^ high capacity column (IBA GmbH) attached to an Äkta Pure system (GE Healthcare). Subsequently, the column was washed until reaching stable A_280_ signal and the protein was eluted in StrepII buffer supplemented with 50 mM biotin (Appendix A).

R-IMAC- and StrepII affinity-pure hZIP1 samples were loaded onto a 24-mL Superdex 200 Increase 10/300 GL column (GE Healthcare) equilibrated with SEC buffer (20 mM Tris-NaOH pH 6, 100 mM NaCl and 10% glycerol) supplemented with 2 mM BME, 0.03% DDM, and 0.003% CHS. Fractions corresponding to the main elution peak were analyzed using SDS-PAGE.

## 3. Results

### 3.1. Overproduction of hZIP-TEV-GFP-His Fusions in S. cerevisiae-Based System

ZIP transporters have proven to be extremely challenging to overproduce and purify in a functional form [30,31]. As the selection of an adequate expression host is a bottleneck in protein production, we decided to investigate whether our established *S. cerevisiae*-based platform will be applicable in manufacturing hZIPs. Based on sequence analysis of the 14 hZIPs, we selected four members possessing the shortest N-termini (Figure 2A), i.e., hZIP1, 2, 11, and 13 with the assumption that the long tails may restrict overall protein rigidity and introduce an additional level of disorder that could be detrimental for the production, stability, and subsequent crystallization-based structure determination approaches. Following the same strategy, we also included the two longest hZIPs, i.e., hZIP6 and 10, as, upon successful production in a stable form, they may still represent the most suitable ZIP family members for structure determination efforts using cryo-electron microscopy (cryo-EM). Thus, this selection of targets widely covers the hZIP family, as these members represent three out of four known subfamilies. Moreover, the selected proteins are related to various pathologies, highlighting their importance as possible drug targets.

The applied *S. cerevisiae* system exploits modified high-copy vectors encoding codon-optimized full-length hZIPs C-terminally fused to tobacco etch virus (TEV) protease-detachable GFP-His-tag, enabling localization, quantification, quality control, and affinity purification (Figure 2B). 

To be able to proceed with downstream studies, high protein yields are crucial. Thus, a satisfactory overproduction level is one of the first milestones to achieve when synthesizing proteins using heterologous systems. In order to assess the initial protein yields, we used the GFP attached to our targets to easily localize and quantify the obtained amounts of recombinant protein. As illustrated by the results from live-cell bioimaging fluorescent micrographs of cells derived from 2-L shaker cultures induced for expression for 48–72 h (Figure 2D), all four short constructs were expressed, and we were able to evaluate the cellular localization of the different targets. However, only minimum traces of protein were obtained for hZIP6 and 10 (data not shown), and hence further work with these members was discontinued. While hZIP1-, hZIP2-, and hZIP11-TEV-GFP-His accumulated mainly in the plasma membrane, hZIP13-TEV-GFP-His localized to intracellular membrane enclosed compartments. This space-varying accumulation in yeast correlates with the natural ability of hZIPs to localize in both plasma and intracellular membranes in human cells. In addition, as reported by the fluorescence signal originating from C-terminally fused GFP, the hZIPs are likely correctly folded and inserted into the yeast membranes. Moreover, by taking advantage of the GFP fusion, we also estimated protein yields per L of small-scale cell cultures (i.e., 2-L following 48–72 h induction). As determined from the whole-cell GFP fluorescence [49] compared with a signal from the GFP standard [41], the anticipated protein yields ranged from 3.3 to 5.7 mg protein per L of cell culture (Figure 2E), as obtained for hZIP1- and hZIP13-TEV-GFP-His, respectively. Such yields are compatible with biophysical, functional, and structural analysis, and thus, they set a promising starting point for further evaluation of the constructs.

Exploiting the ability of correctly folded GFP to exhibit resistance to SDS, we visualized expression of all targets by in-gel GFP fluorescence of whole-cell lysates resolved using SDS-PAGE (Figure 2F). The results demonstrate that the short hZIPs expressed under the above-mentioned conditions accumulated in the yeast membranes in a stable, full-length form, as no apparent fluorescent degradation products were visible. Moreover, the four targets migrated in SDS-PAGE according to their predicted molecular weight, with hZIP13-TEV-GFP-His being the heaviest target (MW of the fusion of 68 kDa, with TEV-GFP-His representing 28 kDa) and hZIP2-TEV-GFP-His being the lightest (62 kDa), respectively. In addition, for all evaluated targets additional weak fluorescent bands of higher molecular mass were observed, indicating presence of possible higher oligomeric states. Based on both the promising production levels and sample quality, we decided to proceed with solubilization screening of all the four short targets.

### 3.2. Detergent Screening and F-SEC Analysis of Solubilized hZIP-TEV-GFP-His Fusions

Prior to most biochemical and biophysical studies, MPs need to be extracted in their native form from the surrounding membranes. Thus, successful purification relies on the capacity of surfactant(s) to efficiently solubilize the membranes and maintain isolated protein stable in solution [52]. To systematically assess and identify detergents that can effectively extract each of the remaining hZIPs, we performed initial screening using a panel of surfactants tested in the presence or absence of cholesteryl hemisuccinate Tris salt (CHS) at a fixed pH value and ionic content (data not shown). In general, the four tested hZIP-TEV-GFP-His fusions appeared difficult to extract in milder surfactants. High solubilization efficacies were achieved only in harsh anionic or zwitterionic detergents (e.g., N-lauroylsarcosine sodium salt or foscholines, respectively), compounds typically not applicable for the downstream investigation of MPs. Three of the examined detergents, i.e., FC-12, a harsh zwitterionic compound, and two milder nonionic surfactants, i.e., DM and DDM, representing the two most commonly used detergents in MP studies [53], displayed favorable extraction efficiencies (Figure 3A). However, both DM and DDM performed only moderately when extracting hZIP2-, hZIP11-, and hZIP13-TEV-GFP-His, yielding solubilization efficacies >10%. Interestingly, for these three detergents, we observed an increase in solubilization efficacy upon supplementation with CHS. However, target-specific variations were observed, with hZIP1-TEV-GFP-His being the most prone and hZIP2-TEV-GFP-His the most refractory to membrane solubilization, respectively. Combining both estimated production levels and extraction efficacies, we decided to proceed with the three most promising candidates, i.e., hZIP1, 2, and 13. In contrast, hZIP11 was excluded due to its low expression levels and inefficient solubilization. Importantly, for the three selected targets, SDS-PAGE analysis of membranes solubilized in all three applied detergents exhibited the expected migration pattern and stability of TEV-GFP-His fusions (Figure 3B), without any visible signs of protein degradation.

In addition to high extraction efficiency, an optimal detergent should preserve the isolated MP in a stable, homogenous form. To assess stability of the extracted hZIP1, 2, and 13, we again took advantage of the TEV-GFP-His-tag and employed F-SEC to evaluate the elution profiles of the detergent-exposed samples (Figure 3C). Thus, for each target, six F-SEC runs were performed and the resulting F-SEC profiles of solubilized fusion proteins were monitored using fluorescence spectroscopy [54]. Overall, the F-SEC data indicate that although all tested hZIPs display some degree of aggregation (as reflected by the peaks eluting at the void volume of the column), the presence of CHS during solubilization significantly improved protein stability. In the case of FC-12, the resulting F-SEC profiles for all three targets were sharp and symmetrical, suggesting monodispersity of all hZIPs in this detergent. However, considering the harshness of this surfactant, as indicated by the highest extraction efficiencies observed in FC-12 (Figure 3A) and large fractions of aggregated protein present in the void, the use of this zwitterionic detergent entails the risk of protein denaturation due to disruption of the protein fold [54,55]. Therefore, we only considered DM and DDM when selecting a primary detergent to obtain hZIPs for downstream applications. Out of three tested targets, hZIP1-TEV-GFP-His clearly displayed the highest stability in both DM and DDM (Figure 3A, top panel), with only marginal fractions of aggregated protein, greatly reduced by CHS. hZIP1 solubilized in DM/DDM + CHS eluted mainly as a sharp, symmetrical peak preceded by a peripheral shoulder (Figure 3C, top panel), suggesting monodispersity of this target. F-SEC peak profiles obtained for the two remaining proteins, i.e., hZIP2 and 13, solubilized in DM/DDM + CHS were enriched in a significant fraction eluting in the void volume (Figure 3C, middle and bottom panels), indicating that a large population of these targets tend to aggregate in these detergents.

All-in-all, hZIP1-TEV-GFP-His exhibited the highest expression levels, extraction efficacy in mild nonionic detergents and post-solubilization stability. Hence, we selected this target for subsequent large-scale purifications in DDM + CHS, representing the most promising surfactant condition for the overproduction of hZIP1.

### 3.3. Purification of the hZIP1-TEV-GFP-His Fusion

hZIP1-TEV-GFP-His was purified from ≈7 g of flask-grown *S. cerevisiae* cells (material obtained from ≈2 L of shaker culture induced for expression for 72 h). Following isolation, crude yeast membranes were solubilized in DDM + CHS (for 2 h at 4 °C; Appendix A) and subjected to immobilized metal ion affinity chromatography (IMAC; Figure 4A). The affinity purification yielded ≈1 mg of protein per L of the cell culture, overall, with relatively high sample purity (Figure 4B). The significant proportion of monomeric form visualized by the SDS-PAGE suggests that, in the applied conditions, hZIP1-TEV-GFP-His was purified mainly as the monomer, but both dimer and high-order oligomers are also visible. Importantly, the produced sample displayed high stability as no fluorescent signal originating from degraded TEV-GFP-His-tag was detected (Figure 4B, left panel). Subsequently, we attempted TEV-GFP-His-tag removal by treatment with TEV protease (Appendix A). However, hZIP1-TEV-GFP-His fusion remained refractory to the enzymatic cleavage, likely due to the single-amino acid length of the C-terminus that results in poor accessibility to the TEV recognition sequence, preventing proteolytic activity.

### 3.4. Design and Purification of N-Terminal hZIP1 Fusions

Following fruitless TEV-GFP-His tag removal from the C-terminal fusion, we attempted design of alternative hZIP1 constructs, omitting GFP and placing the His-tag in either the N- or C-terminus. Initially, we engineered a hZIP1-TEV-His variant to resemble the first studied TEV-GFP-His fusion. However, both low expression levels and extraction efficiencies in nonionic detergents discouraged us from exploiting this construct (data not shown). Therefore, we extended our production strategy with N-terminal hZIP1 fusions. Two selected tags were applied, i.e., His-tag (delivering good expression levels and moderate solubilization properties, as already demonstrated for hZIP1-TEV-GFP-His), and StrepII-tag that has proven useful and convenient in production of other MPs [56], yielding His-TEV-hZIP1 and StrepII-TEV-hZIP1, respectively. After confirming the expression of the two new hZIP1 variants, we performed solubilization screening with increasing concentrations of DDM supplemented with CHS (Appendix A). For His-TEV-hZIP1, the highest solubilization efficacy (≈56%) was obtained for 2% DDM + 0.2% CHS (Appendix A). In contrast, the StrepII-TEV-hZIP1 construct could be completely extracted by all tested concentrations, with the indication that 2% DDM + 0.2% CHS would perform the most efficiently in the large-scale purification (Appendix A). Overall, encouraged by results obtained from this screening, yielding solubilization levels far beyond the C-terminal fusions, we purified both N-terminal hZIP1 variants using single-step affinity chromatography.

Purification of both constructs was performed on crude membranes isolated from ≈40 g of flask-grown *S. cerevisiae* cells derived from ≈12 L of cultures induced for expression for 68 h. Following solubilization of the respective membranes in 2% DDM + 0.2% CHS (Appendix A), either IMAC or StrepII-tag affinity chromatography steps were run (Figure 5). Single-step affinity purification (Figure 5A) yielded ≈0.5 mg of protein per L of the cell culture, however, with moderate sample purity (Figure 5B). Moreover, the analysis of IMAC-pure fractions revealed that His-TEV-hZIP1 was also purified in different oligomeric forms, as reflected by the presence of both monomer and dimer (detectable mainly by immunoblotting; Figure 5B, left panel). Importantly, in contrast to the hZIP1-TEV-GFP-His construct, moving the TEV protease recognition sequence to the N-terminus resulted in marked increase of cleavage efficiency of the His-TEV-hZIP1 variant (Appendix A). Thus, following TEV protease treatment of this N-terminal fusion, tag-free sample of high purity was obtained after reverse-IMAC (R-IMAC) purification step. 

Affinity purification of the StrepII-TEV-hZIP1 construct yielded protein samples of high purity (Figure 5C,D), however, obtained in low amounts (i.e., ≈0.3 mg of protein per L of the cell culture). Thus, in case of both tested N-terminal hZIP1 variants, protein yields were lower than for the hZIP1-TEV-GFP-His construct, perhaps indicative of a stabilizing role of the GFP fusion. Moreover, the presence of an additional band of lower molecular weight in the immunoblot analysis of the samples collected during StrepII-tag affinity purification (Figure 5D, left panel) may suggest either the presence of a contaminant recognized by the antibody or partial protein degradation, the latter supporting a stability enhancing effect of the GFP. Nevertheless, the final purity of the sample can be increased by an additional affinity chromatography step after TEV protease treatment, but this was not attempted here. To assess the stability of produced N-terminal hZIP1 fusions, we proceeded with SEC analysis.

### 3.5. Stability of Affinity-Pure N-Terminal hZIP1 Fusions

In order to assess the quality of the two N-terminal hZIP1 fusions, we analyzed the samples originating from affinity purification by SEC performed in the primary detergent environment, i.e., in the buffer systems containing DDM and CHS (Appendix A). Initially, protein homogeneity was monitored at different pH to enable selection of the most optimal conditions for the subsequent SEC analysis (data not shown). Interestingly, pH scouting revealed that higher sample stability was achieved at lower pH, similarly to what was already reported for hZIP4 [23]. The two hZIP1 variants displayed maximal stability at slightly different pH, i.e., highest homogeneity of His-TEV-free hZIP1 was achieved at pH 6.0 (Figure 6A), whereas the most symmetric SEC profile for StrepII-TEV-hZIP1 was obtained for pH 5.0 (Figure 6B). At the respective optimal pH both hZIP1 variants displayed only marginal aggregation. Moreover, in elution profiles of both hZIP1 samples, the main peak was preceded by a shoulder, hinting at the presence of higher oligomers or partial aggregation in the sample. In addition, as apparent from the SDS-PAGE analysis, SEC enabled the removal of the possible contaminant/degradation product seen in affinity-pure StrepII-TEV-hZIP1 sample. Overall, the generated SEC profiles for both detergent-solubilized N-terminal hZIP1 fusions were symmetric, suggesting high homogeneity and stability of the purified samples. Nevertheless, we observed a gradual decrease of protein stability after repeated rounds of freeze-thawing, as reflected by reduction of the height of the main SEC peak with concomitant increase of aggregated fraction eluting in the void volume (data not shown). Thus, utilization of freshly prepared protein samples should be considered for downstream applications. Finally, the SEC-pure samples exhibited a tendency to precipitate when concentrated above 9 mg mL^−1^, indicating that further optimization of buffer conditions may be required.

## 4. Discussion

The present paper addresses one of the bottlenecks that significantly restricts MP characterization, i.e., the production of prime-quality samples required for their biophysical characterization. Although MPs represent approximately 30% of the human proteome and constitute prominent drug targets [57,58,59], this class of proteins remains poorly understood relative to their soluble counterparts and is highly underrepresented in protein structure databases [60]. Therefore, there is a constant need for fine-tuning of existing heterologous expression systems to deliver recombinant MPs in an economic, efficient, and reliable manner. This dogma applies also to the ZIP family of transporters investigated here, for which no structure of a eukaryotic member has been reported to date. Thus, the overarching aim of this study was to develop an effective expression and purification platform to produce hZIPs. 

Prior to this work, the closest reported attempt of hZIP purification was that of hZIP13 expressed using insect Sf9 cells to investigate the oligomerization state of this target, however, no data on obtained protein yield are presented [24]. In the same publication, the authors also utilized a human 293T cell-based platform, but protein purification was not attempted, and expression was visualized by immunoblotting only. In agreement with this notion, the usage of human cell-derived systems often results in low protein yields and high costs associated with the production [34,61], although strategies to improve expression levels have been proposed [62,63]. Conversely, the *S. cerevisiae* PAP1500 strain-based overproduction system has been employed for isolation of diverse classes of human MPs, and has delivered milligram quantities of samples in a cheap and easy-to-handle manner [39,41,42,43,64]. Importantly, human targets overproduced using this platform recently delivered both X-ray (human aquaporin 10) [39] and cryo-EM (human chloride channel ClC-1) [40] structures. Thus, in this study we took advantage of this protein production pipeline to attempt heterologous overproduction of hZIPs.

To ease downstream screening, all constructs were initially engineered to possess C-terminal TEV protease-detachable GFP-His-tags, enabling detection and affinity purification of the resulting fusions [49,65]. The first overproduction tests of these C-terminally-tagged variants immediately indicated the power of our production strategy, as we were able to recover the four short hZIP transporters selected for this study, all with promising expected expression levels, exceeding 5 mg of protein per L of cell culture in two of the cases, i.e., for hZIP1 and 2 (Figure 2E). On the contrary, both of the longest hZIPs attempted here, i.e., hZIP6 and 10, were refractory to expression using applied strategy. As these targets possess long N-termini (of more than 300 amino acids), it can be speculated that these additional segments may reduce overall protein stability that cannot be rescued by the yeast host. This is consistent with the previous study reporting that even production of an isolated extracellular domain of hZIP4 (with length of 327 amino acids) attempted in *E. coli* was fruitless, as the protein was severely aggregated [28]. Thus, assuming that the full-length hZIP6 and 10 may be more prone to protein degradation than the short hZIP counterparts, and combining it with the C-terminal location of the fusion tag, the detection of any partially translated products would not be possible here. Hence, optimization of the construct design and/or change of the expression host may be necessary for these two targets, but none of these were further explored here.

All four short, C-terminally-tagged hZIPs were synthesized as membrane-embedded full-length proteins. Interestingly, we observed differences in localization of the respective targets in the yeast cells. While hZIP1, 2, and 11 localized largely to the plasma membrane, hZIP13 accumulated intracellularly (Figure 2D), correlating with the hZIP-member specific naturally occurring localization in human cells [10].

As the next step, we sought to identify suitable detergents enabling efficient extraction and stabilizing solubilized hZIPs in their native form. We showed that, in general, the hZIPs investigated here were difficult to extract from the yeast membranes (Figure 3A), regardless of their cellular localization pattern. Based on data from the extensive detergent screening, we can conclude that only harsh zwitterionic detergents of the foscholine family displayed strong solubilization properties, whereas milder nonionic maltosides were rather ineffective, providing extraction efficacy higher than 20% only in the case of hZIP1. This finding agrees largely with the previously observed trend for other classes of MPs expressed in the same host [42,43]. However, the use of zwitterionic detergents entails the risk of protein denaturation leading to the loss of its structure [54,55]. Therefore, even though foscholine-solubilized material exhibited higher symmetry and less void in the F-SEC profiles, DDM was selected as a “safer option” solubilization detergent. Nevertheless, the homogeneity of the DDM-extracted samples was also relatively high, indicating that milder detergents are promising for isolation of hZIPs. Importantly, the presence of CHS increased both the efficiency of solubilization and protein stability. This suggests that this additional presence of cholesterol during solubilization may increase fluidity of the lipid environment of the yeast membranes or the CHS even directly interacts with expressed hZIPs to enhance their stability. The latter, i.e., stabilizing effect of cholesterol, is a well-known phenomenon in the MP field [66].

As the selected construct for initial characterization, i.e., hZIP1-TEV-GFP-His, was refractory to the TEV treatment, likely reflecting poor accessibility of the C-terminus preventing proteolytic activity, design of new constructs was attempted. As the His-tag position is known to contribute to the protein expression levels and stability [67,68], in our strategy we moved the tag to the N-terminus and investigated whether such modification could have beneficial or deleterious effects on the production. Remarkably, the construct comprising an N-terminal His-tag, but devoid of GFP, proved to be TEV-cleavable (Appendix A), but displayed significantly lower purification yields compared with the original C-terminal GFP fusion, although we did not observe major differences in the solubilization efficacy (Appendix A). This observed decrease may suggest lowered levels of expression either as a direct consequence of different location of the tag and/or it may reflect an auxiliary role of the GFP fusion. Indeed, the latter has been reported to preserve the protein fold of other MPs, e.g., G protein-coupled receptors [69]. Thus, extending the N-terminally-tagged hZIP1 construct with GFP may provide more direct possibility of comparing such variant with the C-terminal TEV-GFP-His fusion, but this was not attempted here. Moreover, it cannot be excluded that production of hZIPs can be especially sensitive to the placement of charged tags, such as the His-tag, near the transmembrane domain. Although still not completely understood, it has been shown that cytoplasmic segments of MPs are typically enriched in positively charged residues (mostly lysine and arginine), the so-called “positive-inside rule” [70]. Thus, introduction of an additional positively charged stretch, here in a form of octa-histidine-tag, preceding the noncytoplasmic N-terminal segments of hZIPs may violate the rule, resulting in a lower protein stability. To corroborate this possible effect of the charged tag on production, we included a construct where the His-tag was exchanged to StrepII-tag. StrepII-tag, an octa-amino acid polypeptide, typically does not alter bioactivity of its fusion partner and can be used for affinity purification under mild conditions, maintaining the functionality of produced protein [71]. Interestingly, the production levels of the StrepII-TEV-hZIP1 variant were marginally decreased compared to the His-TEV-hZIP1 construct, again without any detectable changes in detergent extraction efficacy (Appendix A). Collectively, this indicates that the expression of both the N-terminal hZIP1 variants described here may require further optimization of the fusion tag, possibly due to the predicted N_out_-C_out_ topology. Similar conclusions were drawn in a work describing development of purification strategies of bacterial ZIPs sharing the same location of the N- and C-termini, where the insertion of a maltose-binding protein and a signal sequence between the His-tag and the scare N-terminal domain was shown to rescue protein production [32]. 

hZIPs have been shown to form homodimers in a native state, but hZIP6 and 10 can also heterodimerize [13]. Analysis of the in-gel GFP fluorescence of the SDS-PAGE-separated IMAC-pure hZIP1-TEV-GFP-His samples implies that this construct preserves homodimerization to some degree (Figure 4B). The proportion of dimeric versus monomeric form decreases in the N-terminally-tagged GFP-free constructs (Figure 4 and Figure 5), which may correlate with lower stability and/or yields observed for these variants. Hence, while for the StrepII-TEV-hZIP1 construct a small population of the dimeric form was detectable after SEC, hZIP1 (TEV protease-digested His-TEV-hZIP1) eluted mostly as monomer (Figure 6). This may indicate that the His-tag is directly assisting in dimerization, similarly to what was observed for BbZIP, however with the opposing effect [31]. 

Affinity-based protein purification yielded milligram amounts of both N-terminal hZIP1 fusions, although the relative purity of the samples marginally differed (Figure 5). However, while a single chromatography round was sufficient to produce a highly pure sample of StrepII-TEV-hZIP1, two additional steps were included for His-TEV-hZIP1, i.e., TEV protease cleavage and R-IMAC (Appendix A). Although both samples exhibited high monodispersity, as assessed by SEC (Figure 6), the purified variants displayed limited stability, tending to precipitate after freeze–thawing cycles. Moreover, freshly prepared, nonfrozen samples precipitated also at concentrations exceeding 9 mg mL^−1^. However, such concentrations are compatible with requirements for most of the biophysical experiments, including X-ray crystallography [72] and cryo-EM, where even low amounts of the sample can be utilized [73].

In conclusion, we outlined a *S. cerevisiae*-based platform to approach milligram production of hZIPs, with hZIP1 being an example target that was purified to monodispersity. We believe that our strategy for protein expression and purification described here provides a new encouraging option for sample preparation of hZIPs for more detailed biochemical and structural analysis.

## Figures and Tables

**Figure 1 cells-10-00213-f001:**
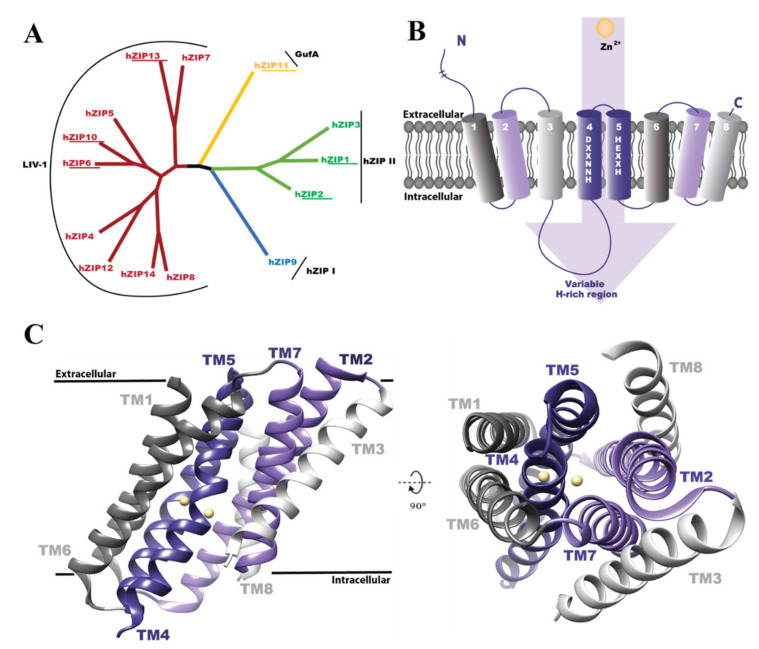
Overview of the Zrt-, Irt-like (ZIP) family of zinc transporters. (**A**) Phylogenetic distribution of the 14 human ZIP transporters (hZIP1-14) classified into four subfamilies, i.e., hZIP I, hZIP II, GufA, and LIV-1. The members investigated in this study are underlined. The phylogenetic tree was constructed using the MEGAX software (https://www.megasoftware.net/) [46]. (**B**) The predicted topology of hZIPs with selected characteristic features indicated. hZIPs likely form eight transmembrane segments (TMs), and the conserved motifs in TM4 and 5 (i.e., HNNXXD and HEXXH, respectively) establish a binuclear metal center. TM3 and 4 are linked by a variable loop containing a histidine-(H)-rich domain. The human ZIPs harbor N-terminal tails of varying length (from ≈6 aa in hZIP2 to ≈407 aa in hZIP10) and a short C-terminus, both oriented towards the extracellular side. (**C**) The only available structure of full-length ZIP from *Bordetella bronchiseptica* (BbZIP) (PDB ID:5TSB) [26,27]. The structure reveals that the TMs arrange pseudo-symmetrically into two domains: TM1, 4, 5, and 6 (dark shades) and TM2, 3, 7, and 8 (light shades). A central binuclear center has been suggested to be involved in Zn^2+^ transfer (yellow spheres) [27]. Side and top views are shown (left and right panels, respectively).

**Figure 2 cells-10-00213-f002:**
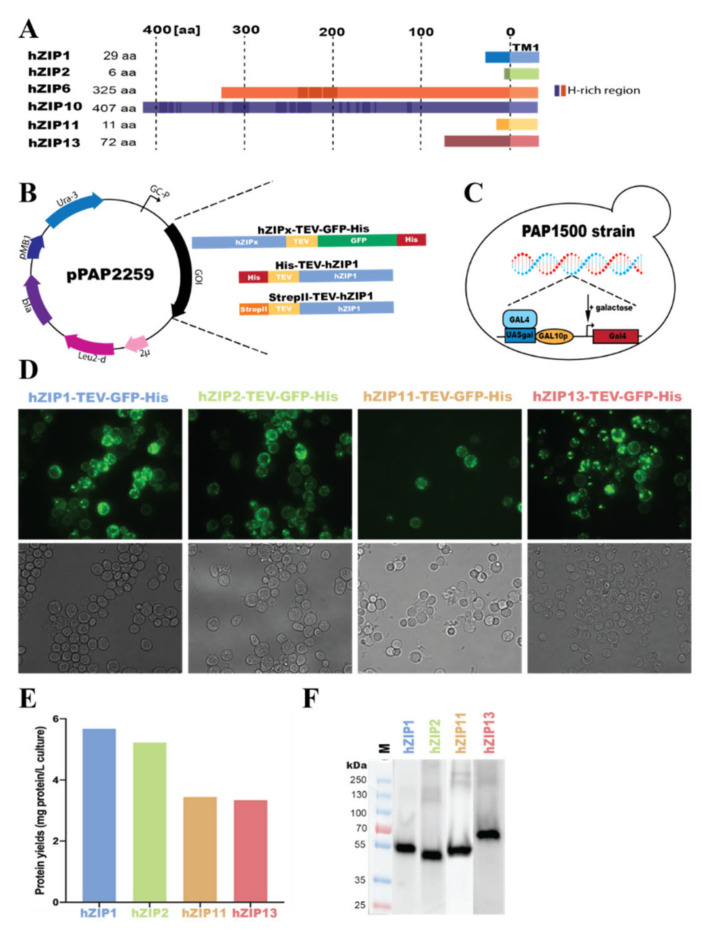
Overproduction of hZIP-TEV-GFP-His fusions in *Saccharomyces cerevisiae*-based system. (**A**) Generalized domain structure of the N-termini of human ZIP targets approached in this study, including the four shortest and the two longest human ZIPs. N-terminal segments of hZIP1 (Q9NY26), hZIP2 (Q9NP94), hZIP6 (Q13433), hZIP10 (Q9ULF5), hZIP11 (Q8N1S5), and hZIP13 (Q96H72) were assigned using TOPCONS topology prediction tool (http://topcons.cbr.su.se) [50]. UniProt (https://www.uniprot.org/) [51] accession numbers of full-length hZIPs protein sequences are shown in brackets. The length of the N-termini, and the positions of histidine-(H)-rich regions (dark boxes) and transmembrane segments TM1 are indicated. (**B**) Map of the *S. cerevisiae* expression plasmid pPAP2259 encoding the different hZIP genes of interest (GOIs) used in the study shown in order of their appearance. Expression constructs were designed to include the full-length forms of the respective hZIPs and different combinations of the following elements: tobacco etch virus (TEV) protease cleavage site, green fluorescent protein (GFP), an octa-histidine- (His) or a StrepII-tag (StrepII). The arrangement of regions of the respective constructs are shown. The expression plasmid also contains a hybrid promotor with the GAL10 upstream activation sequence in the 5′ nontranslated leader of the cytochrome-1 gene (CG-P), a yeast origin of replication (2µ), a β-isopropylmalate dehydrogenase gene with truncated promotor sequence resulting in poor expression of this gene (*leu2-d*), a β-lactamase gene (*bla*), an origin of replication (pMB1), and a yeast orothidine-5-phosphate decarboxylase gene (*URA3*). (**C**) The *S. cerevisiae* protein production strain PAP1500 used in the study. Upon induction with galactose, the strain overexpresses the Gal4 transcriptional activator that is the limiting factor for expression from galactose regulated promoter. GAL10p and UASgal constitute a specific DNA binding site for GAL4 activator. (**D**) Live-cell bioimaging of *S. cerevisiae* cells (derived from 2-L cultures) expressing hZIP1-, hZIP2-, and hZIP13-TEV-GFP-His (72-h induction at 15 °C) or hZIP11-TEV-GFP-His (48-h induction at 15 °C). For each construct GFP fluorescence (top) and differential interference contrast (bottom) micrographs are shown. Magnification: 1000×. (**E**) Estimates of protein production levels of four selected hZIPs expressed as in (**D**). The intensity of the whole-cell GFP fluorescence signal was transformed to the protein amount, by comparison against a free-GFP standard curve fluorescence signal, and the predicted yield is expressed as mg per L of cell culture. (**F**) In-gel GFP fluorescence of SDS-PAGE-resolved whole-cell lysates (i.e., supernatant from centrifuged homogenates) expressing different hZIP-TEV-GFP-His fusions as in (**D**). The predicted MWs of the respective fusions are 63.3 kDa (hZIP1), 61.8 kDa (hZIP2), 63.8 kDa (hZIP11), and 68 kDa (hZIP13). M: marker.

**Figure 3 cells-10-00213-f003:**
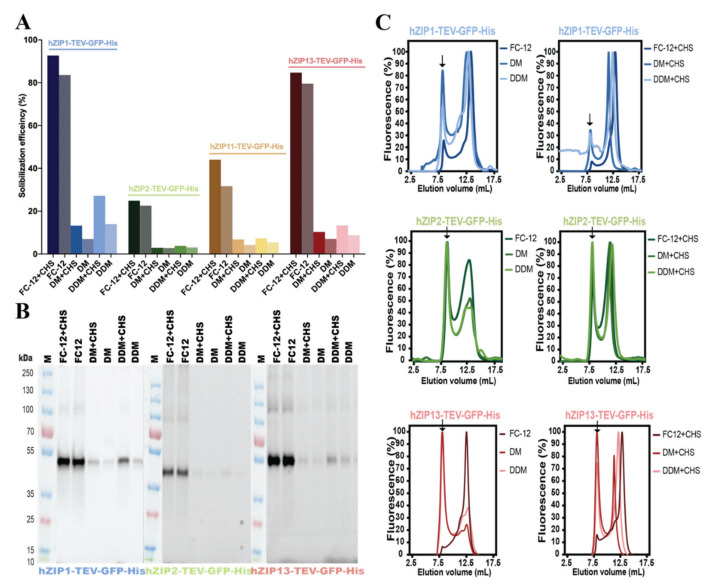
Detergent screening and fluorescence-detection size-exclusion chromatography (F-SEC) analysis of solubilized hZIP-TEV-GFP-His fusions. (**A**) Solubilization efficiency of crude membranes isolated from *Saccharomyces cerevisiae* cells (derived from 2-L cultures) expressing hZIP1-, hZIP2-, and hZIP13-TEV-GFP-His (72-h induction at 15 °C) or hZIP11-TEV-GFP-His (48-h induction at 15 °C). For hZIP1, 2, and 13 analyzed material was solubilized for 2 h at 4 °C with FC-12 (at final concentration of 1%, *w*/*v*), DM (1%, *w*/*v*), and DDM (1%, *w*/*v*) with and without CHS (0.34%, *w*/*v*). For hZIP11, the final concentrations were doubled. Following treatment with the respective detergents, GFP fluorescence was measured in the supernatant after ultracentrifugation and percentage of solubilization efficiency was calculated. CHS: cholesteryl hemisuccinate Tris salt. (**B**) In-gel GFP fluorescence of SDS-PAGE-separated detergent-solubilized crude *S. cerevisiae* membranes overexpressing hZIP1, 2, and 13 C-terminally fused to TEV-GFP-His-tag. Extraction conditions are identical to (**A**) and the analyzed material represents ultracentrifuged supernatant from the respective solubilization. The predicted MWs of the respective fusions are 63.3 kDa (hZIP1), 61.8 kDa (hZIP2), 63.8 kDa (hZIP11), and 68 kDa (hZIP13). M: marker. (**C**) F-SEC analysis of detergent-solubilized crude *S. cerevisiae* membranes overexpressing hZIP1, 2, and 13 C-terminally fused to TEV-GFP-His-tag. Extraction conditions are identical to (**A**) and analyzed material includes ultracentrifuged supernatant from the respective solubilization. Normalized F-SEC chromatograms were obtained after separation on Superdex 200 Increase 10/300 GL column where GFP fluorescence of the eluate was monitored. Arrows indicate the estimated elution positions of the void volume.

**Figure 4 cells-10-00213-f004:**
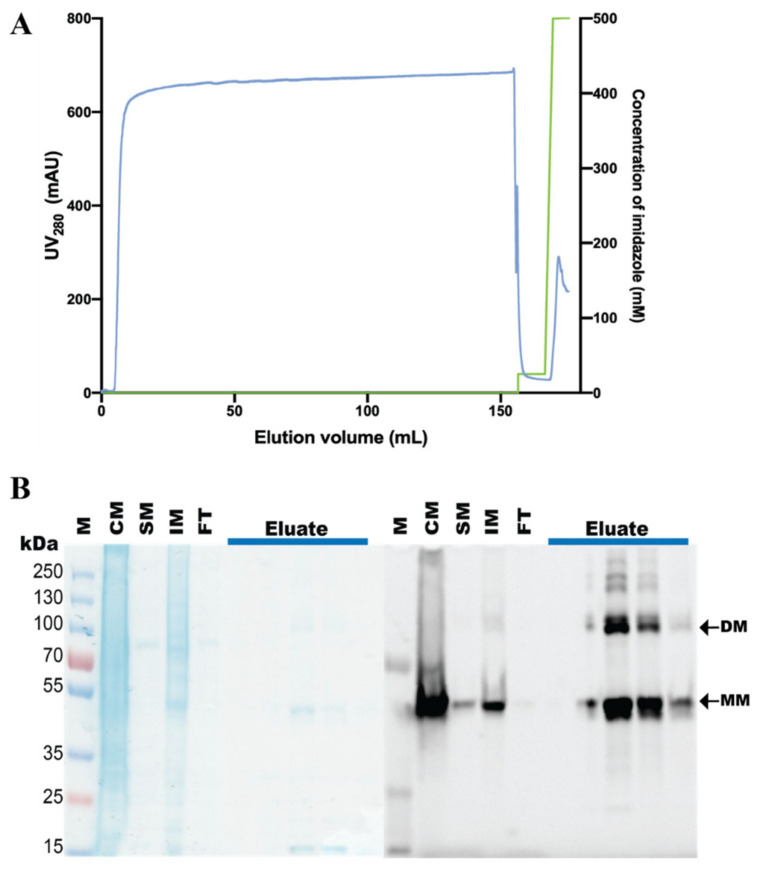
Purification of the hZIP1-TEV-GFP-His fusion. (**A**) Representative purification of hZIP1 C-terminally fused to TEV-GFP-His-tag. Protein was purified from crude *Saccharomyces cerevisiae* membranes isolated from ≈7 g of flask-grown cells and solubilized in 2% (*w*/*v*) DDM + 0.68% (*w*/*v*) CHS for 2 h at 4 °C. Profile from immobilized metal affinity chromatography (IMAC) indicates UV_280_ signal for protein (blue) and concentration of imidazole applied at each step of the affinity purification (green). CHS: cholesteryl hemisuccinate Tris salt. (**B**) Coomassie staining (left) and in-gel GFP fluorescence (right) of the corresponding SDS-PAGE-separated samples collected during IMAC purification. M: marker; CM: crude membranes; SM: solubilized membranes; IM: insoluble material; FT: IMAC flow-through; Eluate: IMAC peak nonconcentrated elution fractions eluted with 500 mM imidazole. Arrows indicate the monomeric (MM) and dimeric (DM) forms of purified hZIP-TEV-GFP-His fusion, with the predicted MWs of the respective fusions of 63.3 kDa (MM) and 126.6 kDa (DM).

**Figure 5 cells-10-00213-f005:**
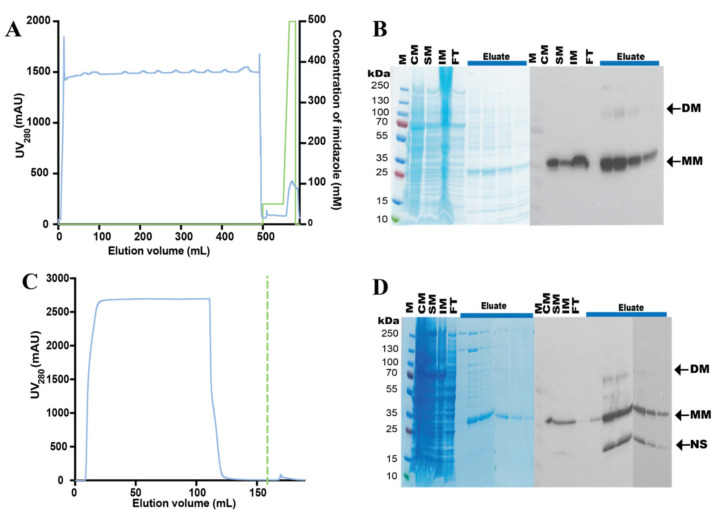
Purification of N-terminal hZIP1 fusions. Crude membranes isolated from *Saccharomyces cerevisiae* cells (derived from 12-L cultures) expressing (**A**,**B**) His-TEV-hZIP1 and (**C**,**D**) StrepII-TEV-hZIP1 (68-h induction at 15 °C) were solubilized in 2% (*w*/*v*) DDM + 0.2% (*w*/*v*) CHS for 2 h at 4 °C. Following solubilization, affinity-based purification was performed. CHS: cholesteryl hemisuccinate Tris salt. (**A**) Immobilized metal affinity chromatography (IMAC) of His-TEV-hZIP1. IMAC profile indicates UV_280_ signal for protein (blue) and concentration of imidazole applied at each step of the affinity purification (green). (**B**) Coomassie staining (left) and immunoblot probed with 6×His mAb-HRP conjugate (right) of SDS-PAGE-separated samples collected during IMAC purification. M: marker; CM: crude membranes; SM: solubilized membranes; IM: insoluble material; FT: IMAC flow-through; Eluate: IMAC peak fractions eluted with imidazole gradient. Arrows indicate the monomeric (MM) and dimeric (DM) forms of purified His-TEV-hZIP fusion, with the predicted MWs of the respective forms of 32.5 kDa (MM) and 65 kDa (DM). (**C**) StrepII-tag affinity chromatography of StrepII-TEV-hZIP1. The profile indicates UV_280_ signal for protein (blue). Green dashed line indicates the starting point of the elution phase with 50 mM biotin. (**D**) Coomassie staining (left) and immunoblot probed with Strep-Tactin^®^ HRP conjugate (right) of SDS-PAGE-separated samples collected during StrepII-tag affinity purification. Letter code of the analyzed samples is to identical to (**B**). Arrows indicate the monomeric (MM) and dimeric (DM) forms of purified StrepII-TEV-hZIP fusion, with the predicted MWs of the respective fusions of 36.5 kDa (MM) and 73.0 kDa (DM). NS: nonspecific band/degradation product.

**Figure 6 cells-10-00213-f006:**
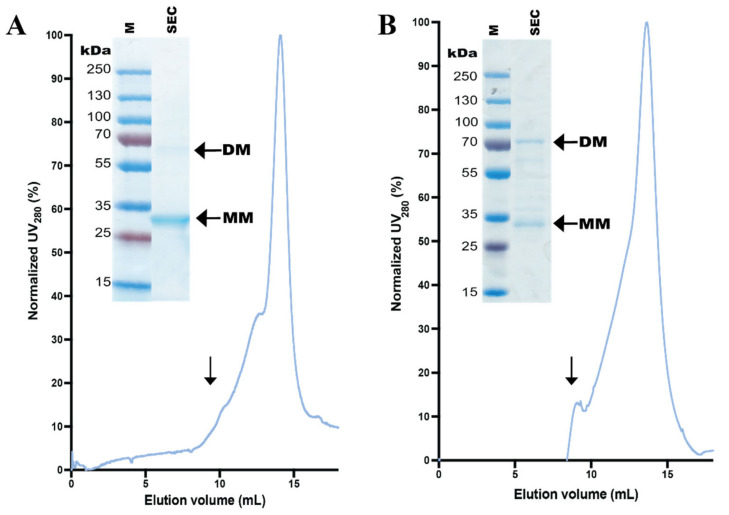
Size-exclusion chromatography (SEC) analysis of N-terminal hZIP1 fusions. (**A**) SEC profile of reverse-immobilized metal ion affinity chromatography-pure hZIP1 (i.e., TEV protease-digested His-TEV-hZIP1). (**B**) SEC profile of StrepII-tag affinity-pure StrepII-TEV-hZIP1. Normalized SEC chromatograms were obtained after separation on Superdex 200 Increase 10/300 GL column where A_280_ was monitored. Arrows indicate the estimated elution positions of the void volume. Insets: Coomassie staining of the resulting SEC-pure SDS-PAGE-separated samples (SEC). M: marker. Arrows indicate the monomeric (MM) and dimeric (DM) forms of the respective N-terminal hZIP1 fusions, with the predicted MWs of His-TEV-free hZIP1 of 32.5 kDa (MM) and 65 kDa (DM), and StrepII-TEV-hZIP1 of 36.5 kDa (MM) and 73 kDa (DM).

## Data Availability

All data presented in this study are contained within this article (Overproduction of Human Zip (SLC39) Zinc Transporters in Saccharomyces Cerevisiae for Biophysical Characterization) and the Appendix A.

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
