# Peer review of "Overproduction of Human Zip (SLC39) Zinc Transporters in Saccharomyces cerevisiae for Biophysical Characterization"

_cells, 2021, doi:10.3390/cells10020213_

Round 1
Reviewer 1 Report
This manuscript describes the use of S. cerevisiae as an expression system for the heterologous expression of human ZIP metal transport proteins. The authors were successful in getting reasonable expression levels and purity for four ZIP proteins. This an important advance because these proteins have been difficult to purify in large quantities from other expression systems.
- Yeast mutants defective for the major transporters for zinc uptake (Zrt1 and Zrt2) are available and it would be simple to determine whether the expressed human transporters are functional in yeast. If their activity could be verified, this would indicate that the expression system produces properly folded and functional proteins for further analysis. This is only a recommendation.
- Line 347. Higher molecular mass forms are visible in the blots for all four proteins and not just hZIP2 and hZIP11.
- Line 359. “rich”, not “reach”
- Line 370. “URA3”, not “Ura-3”
- Line 388. “assess”, not “asses”
- Line 467. It appears to me that hZIP1 was purified mostly as monomer and not as the dimer form.
Author Response
Reviewer’s #1 comments:
This manuscript describes the use of S. cerevisiae as an expression system for the heterologous expression of human ZIP metal transport proteins. The authors were successful in getting reasonable expression levels and purity for four ZIP proteins. This an important advance because these proteins have been difficult to purify in large quantities from other expression systems.
We are pleased to see that the reviewer shares the opinion that our manuscript includes important results that may have a significant impact on the zinc transport field.
- Yeast mutants defective for the major transporters for zinc uptake (Zrt1 and Zrt2) are available and it would be simple to determine whether the expressed human transporters are functional in yeast. If their activity could be verified, this would indicate that the expression system produces properly folded and functional proteins for further analysis. This is only a recommendation.
We thank the reviewer for this excellent suggestion. However, in the present study, we intended to describe employment of our S. cerevisiae-based platform for heterologous expression of human ZIP proteins rather than to report detailed biochemical characterization of these targets. We believe that demonstrating the expression of human ZIPs in yeast together with the ability to extract them from the membranes and affinity purification of a selected target (i.e., hZIP1) represents strong scientific value in itself.
- Line 347. Higher molecular mass forms are visible in the blots for all four proteins and not just hZIP2 and hZIP11.
The reviewer is absolutely correct about this. We have now revised the text in the line 375 to: “In addition, for all evaluated targets additional weak fluorescent bands…”.
- Line 359. “rich”, not “reach”
We thank the reviewer for catching this error. We have now corrected the misspelling in the current line 387.
- Line 370. “URA3”, not “Ura-3”
We thank the reviewer for catching this error. We have now corrected the spelling in the current line 398.
- Line 388. “assess”, not “asses”
We thank the reviewer for catching this error. We have now corrected the misspelling in the current line 429.
- Line 467. It appears to me that hZIP1 was purified mostly as monomer and not as the dimer form.
The reviewer is absolutely correct about this. We have now revised the sentence starting in the current line 510: “The significant proportion of monomeric form visualized by the SDS-PAGE suggests that, in the applied conditions, hZIP1-TEV-GFP-His was purified mainly as the monomer, but both dimer and high-order oligomers are also visible”.
Reviewer 2 Report
The work is very interesting, especially in the respect of methodology, it also has significant practical applications. It confirms the current opinion on the discussed matters. Justification of research topics, selection of the materials and methods, and analysis of the obtained results are correct and appropriate to objective and the discussed issues.
The work is correct from a editorial point, but I have a slight note about the lack of of references to Figs. 2C and 5A.
The methodological part is the strongest element of this work.
Author Response
Reviewer’s #2 comments:
The work is very interesting, especially in the respect of methodology, it also has significant practical applications. It confirms the current opinion on the discussed matters. Justification of research topics, selection of the materials and methods, and analysis of the obtained results are correct and appropriate to objective and the discussed issues. … The methodological part is the strongest element of this work.
We thank the reviewer for critical reading of our manuscript and for providing the feedback. We are pleased to see that the reviewer recognizes the significance of our work, especially its methodological value.
The work is correct from a editorial point, but I have a slight note about the lack of references to Figs. 2C and 5A.
The reviewer is absolutely correct about this. We have now inserted references to the Figure 2C in the current lines 198 and 204 as well as the reference to the Figure 5A in the current line 560.